# Analyzing the impact of sustainable economic development from the policy text network: Based on the practice of China's bay area policy

Huijie Zhou *, Shangjia Yu, Pengyue Wu

College of Science and Technology, Ningbo University, Ningbo, Zhejiang, China

These authors contributed equally to this work.
* zhouhuijie@nbu.edu.cn

**Data Availability Statement:** All relevant data are within https://datadryad.org/stash/share/Ex_AxEs5zeV48h10fsou5AYSLIlyiogTs3LKUKAM1rs or https://doi.org/10.5061/dryad.ffbg79d1s.

## Abstract

In order to break through the surface analysis of the content structure of policy texts, an in-depth discussion of the linkage between regional policy makers and objectives is helpful to analyze the formation mechanism of policy effects. Through social network analysis and multi-index analysis, this study takes the QianwanNew Area of Ningbo and the Guangdong-Hong Kong-Macao Greater Bay Area as representatives to explore the policy framework for the sustainable development of manufacturing industry in the two bay areas respectively. Through the construction of government department cooperation network, policy keyword co-occurrence network, department keyword correlation network, and the analysis of network density, network centrality, structural holes, and cohesive subgroups, it is found that the impact results show great differences, which is related to the network structure of manufacturing policy text.

## 1. Introduction

Text analysis is a data mining technology, mining content includes word frequency statistics, semantic analysis, topic analysis, sentiment analysis, entity recognition, relationship analysis, time series analysis, text classification and text clustering, which is used to discover new and useful information from the original text. In recent years, with the rapid development of the Internet and the arrival of the era of big data, the number of texts is also increasing, and information is mined from a huge number of original texts. More and more scholars begin to focus on machine learning and natural language processing technology to strengthen the ability of text analysis. Major techniques include Naive Bayes(NB) [1], K-Nearest Neighbor(KNN) [2], Support Vector Machines(SVM) [3], econometric methods [4], Conditional Random Fields (CRF) [5], and so on. The quantitative text analysis method based on machine learning provides an empirical and scientific tool for information mining of the texts of Bay Area economic policies and sustainable development policies. For example, through the network analysis of Bay Area sustainable development policy texts, we can understand the network hub nodes of

**Funding:** This work received support from Zhejiang Province Soft Science Research Project (2022C35101) and National Social Science Fund of China(23Bgl013). The funders had no role in study design, data collection and analysis, decision to publish, or preparation of the manuscript.

policy texts, the characteristics of network evolution, and the differences of network structures in different bay areas. In order to better understand the reasons behind the policy and the future development trend, and provide effective suggestions for the implementation of the policy, bring new thinking and application value for the realization of the sustainable development goal of the Bay Area economy. To this end, we call for more, deeper, and more valuable machine learning quantitative text analysis to be put on the agenda in the future.

With an emerging economic form, bay area has become an important carrier for the economic transformation and upgrading of major economies, and its development should meet the requirements of high-quality economic development in China [6–8]. Relying on the Yangtze River Delta Economic Belt, the Qianwan New Area of Ningbois based on a huge manufacturing cluster and industrial base, with advanced manufacturing as the pillar, fully implementing the "Made in China 2025" plan, and actively taking the road of sustainable development of the manufacturing industry. In order to further promote the sustainable development of the manufacturing industry, improve the leadership, support and driving force of regional leading enterprises, especially to promote the sustainable development of the manufacturing industry in the bay area, the central and local governments have issued a package to help the sustainable manufacturing enterprises in the bay area quality development policy.

However, at present, most scholars are more inclined to study the final implementation effect of sustainable manufacturing development policies [9–12], and pay less attention to the formulation and evaluation of policies. "Policies are not too refined." Whether the evaluation of policy effects is effective or not can affect the sustainable development of the regional economy and the national economy to a large extent. At present, the frequency and number of policies related to manufacturing in the bay area have increased significantly, but whether the actual policy effects are synchronized with the requirements for sustainable development remains to be verified.

Therefore, whether the bay area's policy supply for sustainable manufacturing development is effective has become a matter of concern. For example, is the evaluation of manufacturing policies in the bay area represented by the Qianwan New Area of Ningboand the Guangdong-Hong Kong-Macao Greater Bay Areaeffective? What are the differences between the sustainable development policies of manufacturing in the two bay areas? What are the reasons for the differences in policy effects between regions? This paper starts from the supply of manufacturing policies in the bay area, takes the effectiveness of regional policy effects as the starting point of the research, and uses social network analysis to compare the structural characteristics of manufacturing policies in Qianwan New Area and the Guangdong-Hong Kong-Macao Greater Bay Area by mining regional policy texts, to build a cooperation network of policy participating departments, a network of policy keywords, and to evaluate the effect of local policies on the sustainable development of the manufacturing industry in the bay area. This study aims to combine text measurement, multi-index statistical analysis and social network analysis methods to study the current situation of high-quality development of manufacturing in China's bay area from the perspective of social network.

This paper proceeds as follows: Section 2 is a literature review on the effect of industrial policy network and high-quality development of manufacturing industry in the bay area. Section 3 introduces the methodology, and it includes the model specification, policy data sources and design of policy network indicators. Results and discussion can be seen in Section 4–6. Section 4 reports policy network analysis results of the Qianwan New Area of Ningbo and Guangdong-Hong Kong-Macao Bay Area. Comparative analysis of the characteristics of the policy network in the two bay areas will be conducted in Section 5. Section 6 performs evaluation of high quality development system of manufacturing industry in bay area. The related research conclusions and the corresponding policy implications are included in Section 7.

## 2. Literature review

Policy is one of the important factors that affect the development of regional economy, which promotes sustainable development of regional economy by exerting intermediary effects. Domestic and foreign scholars have conducted extensive research on regional industrial policies [13, 14], resource use policies [15], energy investment policies [16–18], R&D investment policies [19, 20], environmental regulation policies [21, 22], and other aspects. These studies have attracted widespread attention from policy makers and provided important references for building a mechanism for sustainable regional economic development. As a new economic form, the economy of the bay area has attracted much attention from the academic and policy circles [23, 24]. A large number of scholars have discussed the industrial policy of the bay area, and some foreign scholars have said that the manufacturing industry creates greater economic value than other industries [25], is an important foundation of the national economy, and policy is the main tool to influence industrial development. Therefore, domestic and foreign experts and scholars have carried out relevant research focusing on the manufacturing industry policy of the bay area. However, after further sorting out the literature, it is found that the current research on industrial policy in the bay area is generally limited to the content and structure of policy texts, without further infiltrating the texts, ignoring the correlation between policy subjects, policy objectives, and policy subjects' influence on policy objectives. There is a lack of analysis of industrial policies in the bay area from the perspective of policy effects. In addition, traditional measurement methods for evaluating a single policy have obvious advantages, but because they do not consider the relationship between policy subjects and policy texts, they are not sufficient to support interpretation when dealing with more complex policy research. For example, the high-quality development of manufacturing in the bay area studied in this paper is a multi-dimensional and complex system, which requires the intervention of a more systematic and structured policy evaluation system, such as the PMC model [26, 27], the social network model, and so on. to explore the interactive relationship between policy subjects and goals [28, 29], or the relationship between micro-individual attributes and macro-social phenomena [30]. Therefore, this research aims to start from the supply of policies in the bay area, introduce social network analysis into the research of government departments-policy keywords, and explore the cooperative relationship between the main bodies of local policy formulation, the focus of content, and the adaptability between the main body and the content. The differences in the policy texts of the Qianwan New Area and the Guangdong-Hong Kong-Macao Greater Bay Area, based on the current situation of the bay area, provide reference and suggestions for the optimization of the sustainable development policy of the manufacturing industry in China's bay area.

In addition, domestic and foreign experts and scholars have also conducted multi-perspective discussions on the economic development policy of the bay area, and found that there is a lack of relevant policy research reviews on the bay area economy in domestic and foreign academic circles [6]. Judging from the existing literature, most domestic experts and scholars have discussed the status quo of industrial development in the bay area and put forward corresponding improvement measures. However, there is a lack of systematic literature review from the industrial policy level, especially for the special geographical characteristics of the bay area. Most of the research results focus on the strategic positioning of industrial development in the bay area, development measures and other planning and design and public policies, and rarely involve basic research policies such as the spatial structure characteristics and development laws of the bay area regional economy [31–34].

After combing the literature on industrial policy in the bay area in the past ten years, the representative researches are as follows: Ning L and Xu L [35] used the theoretical framework

of new regionalism as a research tool to explain from the public policy level to the industry in the Beibu Gulf Economic Zone. Policy conflicts have led to the convergence of regional industrial structures, and emphasis should be placed on improving the soft environment for industrial policies. Similarly, Wu J [36] employed the SVAR method and combined with the exogenous shock of economic policy uncertainty, proposed that the shock of industrial policy uncertainty will have a certain negative impact on the industrial economy of the bay area. Zhang L and Liu J [34] found that the Guangdong-Hong Kong-Macao Greater Bay Area has problems such as low matching between policy tools and goals, and unbalanced policy allocation resources according to the research structure among policy subjects, tools and goals. It also proposes an organic combination of industrial policy tools, dynamically adjusts the industrial policy system, and simultaneously improves supporting policies to coordinate resource allocation between regions. Wang M and Cai Z [37] pointed out that the bay area policy must have a clear industrial division of labor, and strengthen policy pertinence to achieve industrial dislocation development and exert agglomeration effect. At the same time, policy evaluation is based on the collected actual implementation effects and benefits of policies, and conducts causal analysis of policy objectives and policy effects through scientific analysis tools. Scientific policy evaluation not only provides a rational basis for decision makers, but also helps policy entities to accurately analyze and adjust in time in each implementation stage, laying a good foundation for future policy application and practice [38]. For this reason, domestic scholars believe that policy evaluation can mainly be studied from the aspects of value judgment, evaluation theory, evaluation standard and index system, evaluation subject, evaluation method and tools [39–42]. Interms of manufacturing industrial policy, Liu Let al. [43] evaluated the impact of high-end equipment manufacturing innovation policies by building a three-dimensional policy (policy strength, number of issuing departments, and frequency) quantitative model. Wang L and Wang S [44] used the entropy method and the data envelement method to conduct an empirical analysis on the implementation effect of the sustainable development policy of the high-tech industry in the Yangtze River Delta region.

## 3. Methodology

As a specific graph theory method [45], social network analysis is mainly used to evaluate the quality of each stakeholder (research subject) to obtain variables that affect their development activities [46–48]), especially focusing on whether and what kind of association exists between variables [49], therefore, is widely used in the field of science.

As a special regional economy, the Bay Area has the policy effect of identifying and utilizing the knowledge in the policy texts and the related information between the knowledge. The specific steps to build the high-quality development policy text network for the manufacturing industry in the Bay Area are as follows.

Step first, policy text collection. Collection of policy texts refers to the process of collecting policy texts from various sources, including official government websites, news media and existing data sets [50]. This study conducted an empirical study on the high-quality development policy documents of the manufacturing industry issued by the administrative departments of Qianwan New Area and Guangdong-Hong Kong- Macao Greater Bay Area. In the collection of relevant policy texts, "Bay Area" and "manufacturing industry" were first taken as keywords, and then a Pathon crawler program based on request database was compiled. Finally, we logged in to the official websites of the administrative departments of Qianwan New Area and Guangdong- Hong Kong-Macao Greater Bay Area, and used the crawler to crawl the information under the policy section, including policy title, release time, release agency, and full text of the policy.

**Table 1. List of policy text data of two bay areas.**

| | Time | Number of Policy Texts | Number of Issuing Departments | |
|---|---|---|---|---|
| Qianwan New Area | 2016 | 1 | 1 | Region, service, competitiveness, R & D, foreign trade, transformation and upgrading, etc |
| | 2017 | 2 | 3 | Competitiveness, characteristics, clusters, innovation, science and technology, cultivation, etc |
| | 2018 | 2 | 6 | Quality, intelligence, reform, innovation, transformation and upgrading, green, etc |
| | 2020 | 3 | 8 | Quality, technology, cultivation, environment, green, brand, etc |
| | 2021 | 1 | 1 | Green, production, service, resources, clusters, technology, etc |
| Guangdong-Hong Kong-Macao Bay Area | 2009 | 1 | 1 | Equipment, market, cultivation, capital, technology, brand, etc |
| | 2012 | 1 | 3 | Industry, innovation, scale, technology, services, resources, etc |
| | 2014 | 1 | 1 | Clusters, new energy, funds, subsidies, taxes, services, etc |
| | 2016 | 1 | 10 | Internet, innovation, green, sharing, reform, intelligence, etc |
| | 2017 | 1 | 2 | Taxation, land, reform, resources, green, talents, etc |
| | 2018 | 1 | 2 | Internet, industrial chain, digitization, cluster, ecology, finance, etc |
| | 2019 | 1 | 14 | Ecology, trade, science and technology, innovation, finance, transformation and upgrading, etc |
| | 2020 | 2 | 2 | Digitization, networking, technology, platform, equipment, cluster, etc |
| | 2021 | 4 | 8 | Innovation, value chain, competitiveness, environment, market, green, etc |

Step two, data preprocessing. The collected policy text is preprocessed, such as using regular expressions to realize Chinese text clauses, deleting illegal characters, and manually marking policy entities, so as to facilitate the follow-up research.

Step three, policy entity identification based on deep learning. Natural language processing knowledge and named entity recognition technology are used to dig deeply into policy texts, and key knowledge entities, such as government agencies and mentioned policies, as well as co-occurrence and inclusion relationships among entities, are identified from policy texts through semantic annotation and entity extraction, so as to achieve the goal of extracting structured knowledge from unstructured policy texts. The time span of the collected policy texts is 20 From January 2009 to December 2021, the government departments of the two major bay areas searched for manufacturing policies, and collected 22 major policy texts in the two major bay areas, as shown in Table 1.

Due to the late proposal of the construction of the bay area in China and the late start of the construction of the manufacturing industry in the bay area, generally speaking, there are not many manufacturing policy documents in the Qianwan New Area, and only 9 manufacturing policy documents have been collected. Among them, the Ningbo Qianwan New Area Management Committee (formerly the Hangzhou Bay Area Development and Construction Management Committee) has formulated the most policies, which matches its authority to lead the economic development of the manufacturing industry in the bay area. The number of policy texts is the largest, and the number of other government departments in the Guangdong-Hong Kong-Macao Greater Bay Area, whether formulated alone, jointly or with participation, is more than that of the Qianwan New Area.

Step four, the construction of policy network model. Policy network model is a research method to analyze the interaction between diversified policy subjects in the policy process [51]. This study is based on the dependency multi-layer network proposed by Buldyrev S et al. [52], and draws on the dependency network under intergovernmental relations [53], which organically combines the departmental cooperation network and the keyword co-occurrence

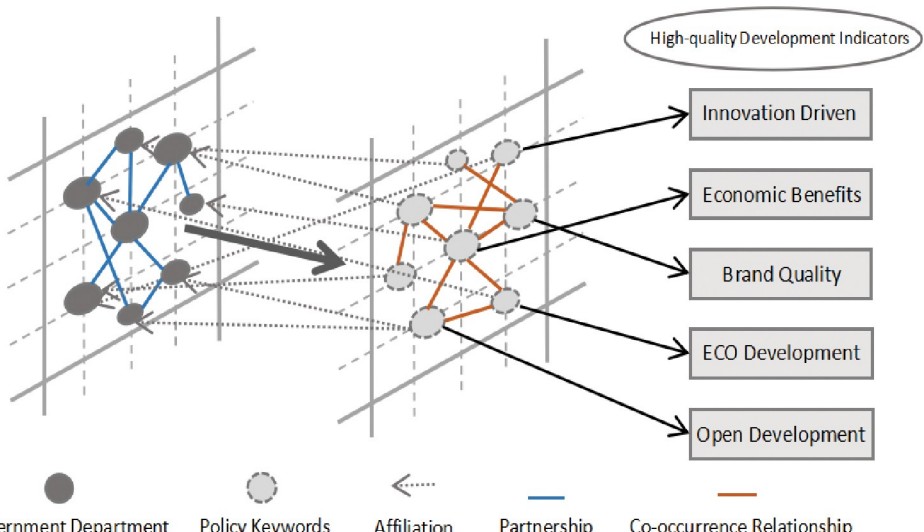

**Fig 1. Design map of manufacturing sustainable development policy network.**

network. The multi-layer network consists of more than two single networks, breaking through the restriction of homogeneity of nodes and edgeways, and including the membership network between nodes of different nature, which is organically combined with the cooperation network of government departments and the co-keyword network of high-quality development policy of manufacturing industry. Based on this, the correlation values among policy texts are quantitatively calculated to analyze the participation of Bay Area government departments in the policy formulation process and the focus of policy development. The policy network model constructed by the research is shown in Fig 1.

Step five, policy network index design. This paper conducts complex social network analysis for the manufacturing policies of Qianwan New Area and Guangdong-Hong Kong-Macao Greater Bay Area. Common network analysis metrics in this method are selected, including centrality index, structural hole index, network density index, etc. The definition and calculation methods of each index are as follows.

## (1) Centrality index

In the departmental network, the point centrality represents the number of connections between nodes; in the keyword network, it can explain the attention of policy texts to policy goals. Betweenness centrality represents the degree of influence of the node on other nodes in the departmental network; in the keyword network, it represents the degree of influence of the keyword on the co-occurrence of other keywords.

## (2) Index of structural holes

Structural holes represent the degree of control that the node has over the information transmission capabilities of other nodes. In policy network analysis, effective scale and restriction are often selected for analysis.

Effective scale represents the possibility of structural holes in nodes in the network, and its value is negatively correlated with redundancy, the closer it is to the core of the network. In the departmental network, the degree of influence of the department on other departments in the process of policy formulation can be measured; in the keyword network, it is shown as the

core position of the keyword, which is used to explain the coordination and consistency of the content of the policy text and the goal; direct or indirect relationship between nodes in the network. The smaller the value, the smaller the constraint, the greater the possibility of obtaining non-redundant information, the more likely it is to be close to the core of the network, and the more likely it will occupy structural holes [54].

## (3) Network density indicators

Network density can be used to describe the density of interconnected edges between nodes in the network, also known as agglomeration. It can be described as in the department and keyword network, if there is a relationship between A and B and C, and there is also a relationship between B and C, then the network density can be used to measure the closeness between the three.

## (4) Group centrality and group intermediation

Group Centrality and group intermediation are mainly used to describe the overall characteristics of departments and keyword networks. The calculation formulas are as follows:

Group Centrality:

$$G_n = \sum_i^n [DC(i)^* - DC(i)] / \max_n [DC(i)^* - DC(i)] \tag{1}$$

Among them, $DC(i)$ represents the total number of connections between node $i$ and other nodes $j$, that is, $DC(i)$.

Group Intermediation:

$$G_b = 2\sum_i^n [p_{jk}(i)^* - p_{jk}(i)] / [(n-1)^2 - (n-2)] \tag{2}$$

Among them, $p_{jk}(i)$ represents the ratio of the shortest path through node $i$ to all paths through node $i$, namely $p_{jk}(i)$ (represents the number of shortest paths between nodes $j$ and $k$, and represents the number of shortest paths through node $i$).

## (5) Evaluation indicators for sustainable development

Since the actual meanings of the selected sustainable development indicators are different, their importance in the comprehensive evaluation indicators is also different, that is, the weight of each indicator is different. Therefore, in this paper, the evaluation index is firstly processed by the range method without dimension, and the calculation formula is as follows:

$$x_i^* = \frac{x_i - \min_n(x_i)}{\max_n(x_i) - \min_n(x_i)} \tag{3}$$

Then calculate the dispersion coefficient of each index (the ratio of mean $x_i$ to standard deviation $\delta_i$), and then the index weight can be obtained according to the proportion of the dispersion coefficient. The calculation formula is as follows:

$$Q_i = \sum \left( \frac{\delta_i}{X_i} \middle/ \sum_i^n \frac{\delta_i}{X_i} \right) x_i \tag{4}$$

Step six, policy text correlation calculation. The policy text is calculated based on the dependency multi-layer network under the established inter-governmental relationship. This study

uses the social network analysis method to explore the characteristics of high-quality development policy networks of the manufacturing industry in Qianwan New Area and Guangdong-Hong Kong-Macao Greater Bay Area from the perspectives of cooperation among policy participating departments, co-occurrence of policy keywords, comparison of overall cooperation closeness, comparison of keyword network characteristics, and correlation between departments and keywords.

# 4. Policy network analysis of two bay areas

## 4.1 Policy network analysis of the Qianwan New Area of Ningbo

According to the department cooperation frequency of QianwanNew Area, the department cooperation matrix shown in Table 2 is obtained. And use gephi software to draw the department cooperation network diagram of QianwanNew Area from 2009 to 2021, as shown in Fig 2. The node size in the figure indicates the degree of cooperation among departments; The thickness of the connection depends on the number of collaboration between departments. (Please refer to S1 Appendix at the end of the text for the names of government departments represented by the letters in the matrix).

It can be seen from the figure that from 2009 to 2021, 11 departments participated in the manufacturing policy-making of Qianwan New Area. The development and Construction Management Committee of Ningbo Qianwan New Area has the most policy-making times and is in the central position in the network, reflecting its closest cooperation with other departments, which is consistent with the Department functions of the Management Committee of the new area and its goal of promoting the high-energy development of industrial agglomeration.

**4.1.1 Analysis of cooperation between policy participating departments.** *(1) Centrality analysis.* According to the position of government departments in the cooperation network, use UCINET6.0 to calculate the centrality index of each department, and use SPSS to draw the scatter diagram for visual processing, as shown in Fig 3. The numbers in the figure correspond to the serial numbers of government departments in Table 3.

When the node is in the first quadrant, it indicates that the Department represented by the node cooperates with other departments in the network for many times and has a close relationship. The four departments of the development and Construction Management Committee of Hangzhou Bay New Area, the Bureau of Commerce, the Economic Development Bureau and the municipal quality and technical supervision branch are all in the first quadrant, indicating that they have strong control over resource coordination. The Management

**Table 2. Department cooperation matrix of QianwanNew Area.**

| Department | A | B | C | D | E | F | G | H | I | J |
|---|---|---|---|---|---|---|---|---|---|---|
| A | **0** | 2 | 1 | 2 | 1 | 2 | 1 | 1 | 1 | 1 |
| B | 2 | **0** | 1 | 1 | 1 | 1 | 1 | 0 | 0 | 0 |
| C | 1 | 1 | **0** | 0 | 0 | 0 | 0 | 0 | 0 | 0 |
| D | 2 | 1 | 0 | **0** | 1 | 1 | 1 | 1 | 1 | 1 |
| E | 1 | 1 | 0 | 1 | **0** | 1 | 1 | 1 | 1 | 0 |
| F | 2 | 1 | 0 | 1 | 1 | **0** | 1 | 0 | 0 | 1 |
| G | 1 | 1 | 0 | 1 | 1 | 1 | **0** | 0 | 0 | 0 |
| H | 1 | 0 | 0 | 1 | 1 | 0 | 0 | **0** | 1 | 1 |
| I | 1 | 0 | 0 | 1 | 1 | 0 | 0 | 1 | **0** | 1 |
| J | 1 | 0 | 0 | 1 | 0 | 1 | 0 | 1 | 1 | **0** |

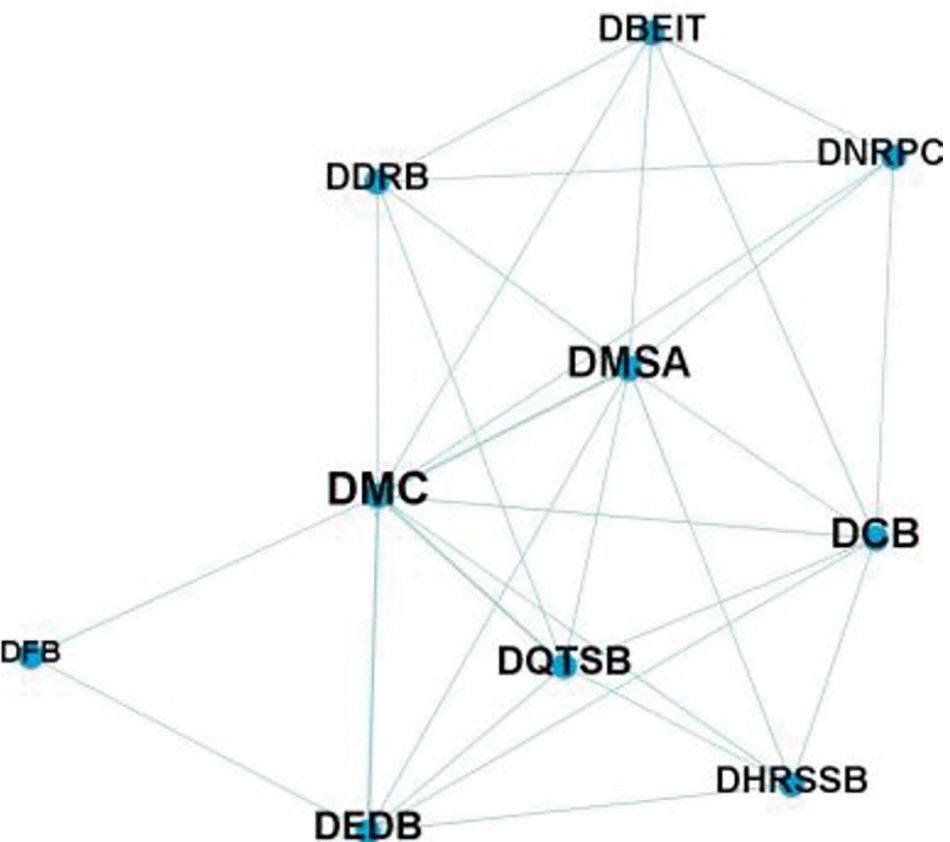

**Fig 2. Department cooperation network of QianwanNew Area.**

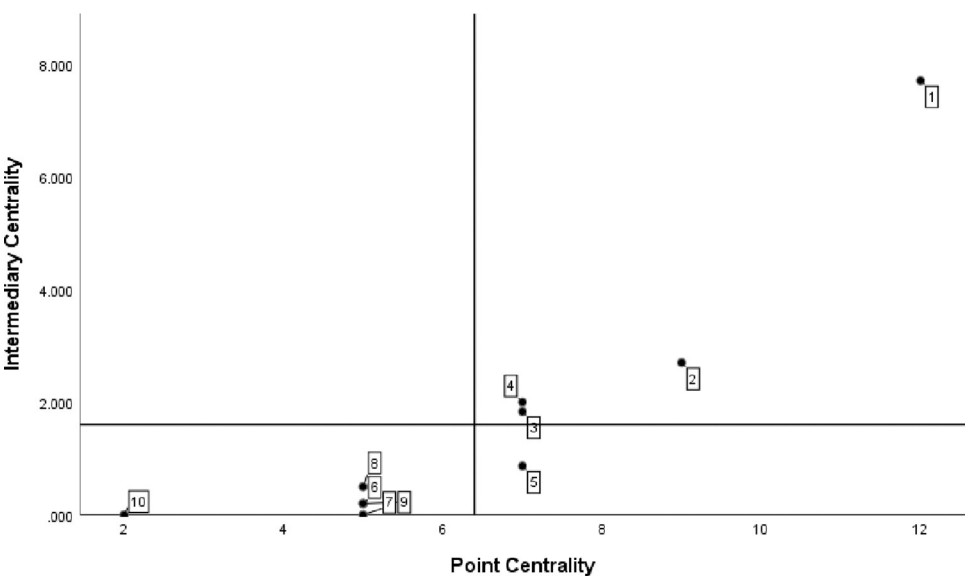

**Fig 3. Analysis of centrality of government departments in QianwanNew Area.**

**Table 3. Structural indicators of government.**

| Serial Number | Department | Effective Scale | Limit Degree |
|:---:|:---:|:---:|:---:|
| 1 | DMC | **5.208** | **0.374** |
| 2 | DMSA | **3.667** | **0.449** |
| 3 | DBC | **3.714** | **0.497** |
| 4 | DEDB | 2.786 | 0.581 |
| 5 | DQTSB | 2.571 | 0.583 |
| 6 | DBEIT | 2 | 0.653 |
| 7 | DNRPCB | 2 | 0.653 |
| 8 | DDRB | 2.4 | 0.662 |
| 9 | DHRSSSB | 2 | 0.681 |
| 10 | DFB | 1 | 1.389 |

Committee of the new area is mainly responsible for the regional economic and social development and construction planning, and is at the helm of the development of manufacturing industry in the bay area. The Commerce Bureau, the Economic Development Bureau and the branch of the Municipal Bureau of quality supervision are responsible for industrial investment, technological innovation, enterprise cultivation and green manufacturing in the new area, which is the second main force for the development of manufacturing industry in the bay area; The node is in the third quadrant, indicating that the representative Department of the node is located at the edge of the cooperation network, has low contribution rate to policy-making, and has relatively weak power to coordinate and control manufacturing resources. The human resources and Social Security Bureau and the Finance Bureau of Hangzhou Bay New Area are in the third quadrant. According to the function description of the two institutions, the guess reasons are as follows: the main responsibility of the human resources and Social Security Bureau of Hangzhou bay new area is to coordinate and lead the regional human resources development, employment management and talent introduction, which is the threshold institution for talent introduction and talent security of manufacturing enterprises; Its strong independence results in relatively little contact with other departments and institutions; The finance bureau is mainly responsible for finance, audit, asset management and other work. It generally plays a cooperative role in the process of manufacturing policy-making, that is, other highly relevant departments formulate policies, and the finance bureau is responsible for financial allocation and fund management according to functional requirements. Therefore, it is not close in the departmental cooperation network.

*(2) Structural hole analysis*. Use UCINET6.0 to get the index of departmental structure hole, as shown in Table 3. Based on the two indicators of effective scale and restriction degree, the effective scale and restriction degree of Ningbo Qianwan New Area Management Committee, market supervision and Administration Bureau and Commerce Bureau are higher, indicating that these three departments play an important role in resource coordination in the cooperation network of government departments in QianwanNew Area. In addition, the Economic Development Bureau of Hangzhou Bay New Area has undertaken the responsibilities of innovative development, agglomeration development and export-oriented development of small and micro enterprises, including the manufacturing industry; The Municipal Bureau of quality and technical supervision, in collaboration with the bay area Economic Development Bureau, undertakes the function of innovation support and opens resources such as scientific research experiments for small and micro enterprises in the manufacturing industry. As the main departments of the sustainable development of manufacturing industry in the bay area, the Bureau of economy and information technology and the Bureau of natural resources,

planning and construction have small effective scale and high restriction, indicating the need to strengthen their position and role in the sustainable development of manufacturing industry and their resource coordination ability. (Please refer to S1 Appendix at the end of the text for the names of government departments represented by the letters in the matrix).

**4.1.2 Co-occurrence analysis of policy keywords.** In social network analysis, common words network analysis is the most common analysis method in literature analysis. In the network diagram, nodes refer to keywords, the size of nodes indicates the status of keywords, the connection of nodes indicates the connection between keywords, and the number of connections represents the co-occurrence times of key words in the same text. According to whether the keywords appear in the same text, use Net draw software to draw, as shown in Fig 4. Keywords such as subsidies, medical care, Internet, agglomeration and services are in the core position in the co-occurrence network, followed by transformation and upgrading, competitiveness, industrial chain, education, etc. It shows that in order to effectively promote the transformation, upgrading, quality improvement and innovative development of industrial economy in the bay area, QianwanNew Area promotes the agglomeration of industrial clusters by giving special subsidies to medical device industry, manufacturing service industry and other special funds.

*(1) Centrality analysis.* The scatter diagram of policy keywords is shown in Fig 5, and the number label in the figure is the same as that in Table 4. From the distribution density of the

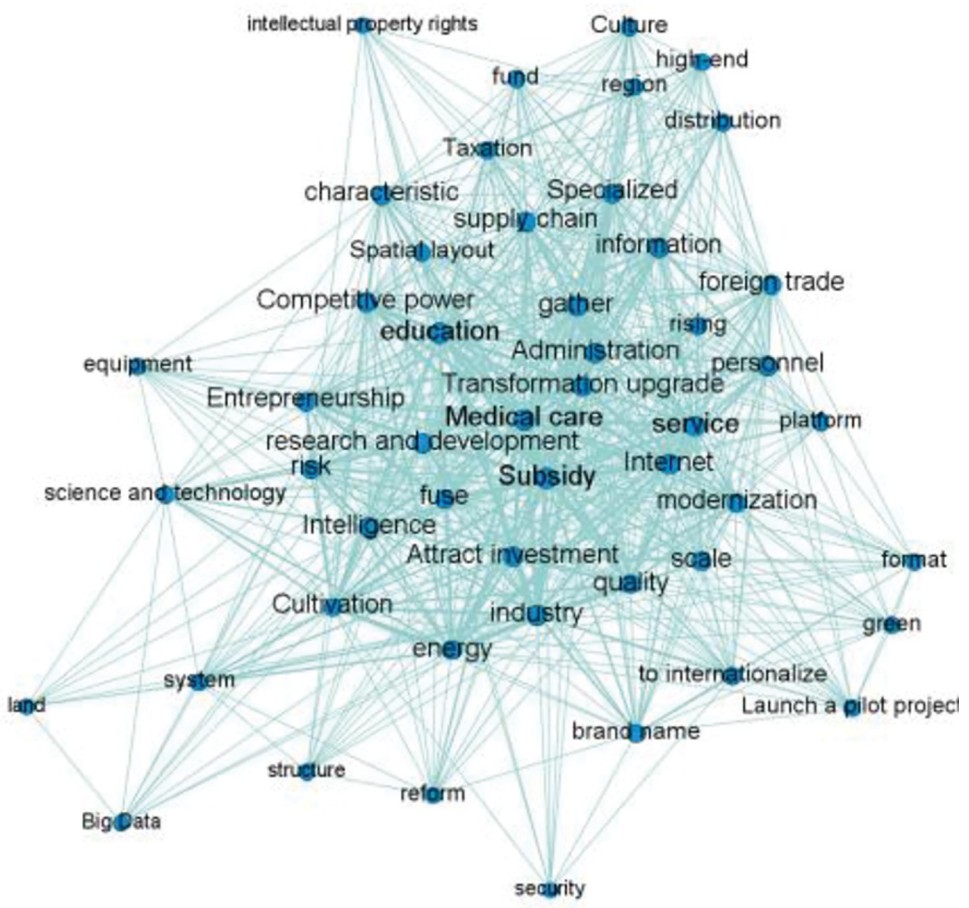

**Fig 4. Co-occurrence network of keywords of manufacturing policy in QianwanNew Area.**

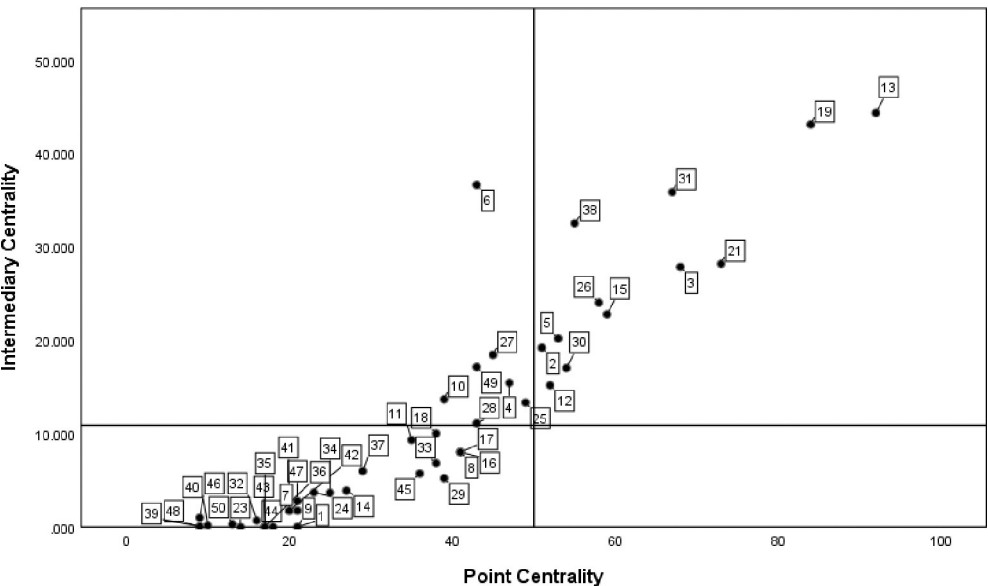

**Fig 5. Value chart of policy keyword point centrality and intermediary centrality.**

digital label quadrant, it can be seen that the number of core keywords in the keyword co-occurrence network is small, and the keywords such as subsidies, medical treatment and services are located in the first quadrant, indicating that such keywords are the development focus or research hotspot in the policy text. Government subsidies, basic service facilities and

**Table 4. Structural indicators of policy keywords in QianwanNew Area.**

| Serial Number | Key Word | Effective Scale | Limit degree | Serial Number | Key word | Effective Scale | Limit Degree | Serial Number | Key Word | Effective Scale | Limit Degree |
|---|---|---|---|---|---|---|---|---|---|---|---|
| 1 | Region | 7.716 | 0.186 | 18 | Personnel | 15.497 | 0.133 | 35 | Format | 6.607 | 0.23 |
| 2 | Transformation and Upgrading | 20.171 | 0.119 | 19 | Medical care | 26.358 | **0.102** | 36 | Green | 6.607 | 0.23 |
| 3 | Service | 23.377 | **0.107** | 20 | Fund | 7.716 | 0.186 | 37 | Brand | 12.774 | 0.168 |
| 4 | Competitive power | 18.239 | 0.129 | 21 | Education | 22.621 | 0.112 | 38 | Attract investment | 21.72 | 0.122 |
| 5 | Administration | 20.307 | 0.117 | 22 | High-end | 7.716 | 0.186 | 39 | Land | 3.843 | 0.417 |
| 6 | Research and Development | 22.253 | 0.116 | 23 | Equipment | 6.307 | 0.278 | 40 | Big data | 4.408 | 0.379 |
| 7 | Logistics | 8.943 | 0.179 | 24 | Science and Technology | 9.733 | 0.216 | 41 | Rising | 9.571 | 0.188 |
| 8 | Information | 15.909 | 0.13 | 25 | Cultivation | 16.291 | 0.154 | 42 | Spatial distribution | 9.571 | 0.188 |
| 9 | Culture | 7.716 | 0.186 | 26 | Energy | 19.167 | 0.137 | 43 | Experimental unit | 6.607 | 0.23 |
| 10 | Foreign trade | 16.611 | 0.13 | 27 | Intelligence | 19.045 | 0.129 | 44 | Platform | 7.086 | 0.218 |
| 11 | Characteristic | 14.952 | 0.143 | 28 | Fuse | 16.716 | 0.138 | 45 | Scale | 13.976 | 0.147 |
| 12 | Gather | 18.318 | 0.123 | 29 | Entrepreneurship | 14.042 | 0.149 | 46 | Intellectual property right | 5.962 | 0.349 |
| 13 | Subsidy | 26.581 | **0.102** | 30 | Quality | 18.976 | 0.13 | 47 | System | 8.26 | 0.244 |
| 14 | Tax | 10.87 | 0.165 | 31 | Industry | 23.07 | 0.116 | 48 | Security | 3.861 | 0.422 |
| 15 | Internet | 21.889 | 0.115 | 32 | Reform | 6.508 | 0.269 | 49 | Risk | 17.684 | 0.135 |
| 16 | Informatization | 20.856 | 0.129 | 33 | Internet | 18.548 | 0.141 | 50 | Structure | 6.19 | 0.301 |
| 17 | Logistics | 15.416 | 0.153 | 34 | Share | 6.23 | 0.283 | | | | |

medical devices are the resources and direction of the development of manufacturing industry, which is of great practical significance to promote the sustainable development of manufacturing industry. Intelligence, R & D, foreign trade and competitiveness are powerful aids to promote the development of manufacturing industry. Policy keywords such as land, big data, intellectual property and security are in the third quadrant, indicating that they have a weak impact on the realization of policy effects or policy support.

*(2) Structural hole analysis.* The structural indicators of policy keywords are shown in Table 3. It can be seen that subsidies, medical treatment and services have the lowest restrictions, and the effective scale ranks in the top three from large to small. It shows that such keywords are less affected by the changes of policy subjects in the policy text, and have certain practical significance. However, security, land, big data and intellectual property have high restrictions and low effective scale, indicating that such keywords have high redundancy, so they play a weak intermediary role in the process of information transmission and are easy to be affected by the change of policy subjects.

*(3) Condensed subgroup analysis.* Condensed subgroup analysis is a common and important method in social cooperation network analysis to explore cluster phenomenon. This paper uses UCINET6.0 The Concor algorithm (iterative correlation convergence method) in 0 visualizes the characteristics of the policy keyword network and obtains the keyword aggregation subgroup classification results, as shown in Fig 6.

The analysis shows that the key words of manufacturing policy in the new area are divided into 8 agglomerative subgroups with different numbers, which are co-polymerized into 4 categories. The first category mainly reflects the two dimensions of open development and innovation driven. Represented by logistics, culture, foreign trade, taxation, transformation and upgrading, competitiveness, Internet and talents, it reflects the need for sustainable development of manufacturing industry in the bay area to coordinate multi industrial factors and improve competitiveness; The second category mainly reflects the two dimensions of economic benefits and brand quality, represented by R & D, education, service, management, agglomeration, industrial chain and specialization, reflects the requirements of the bay area

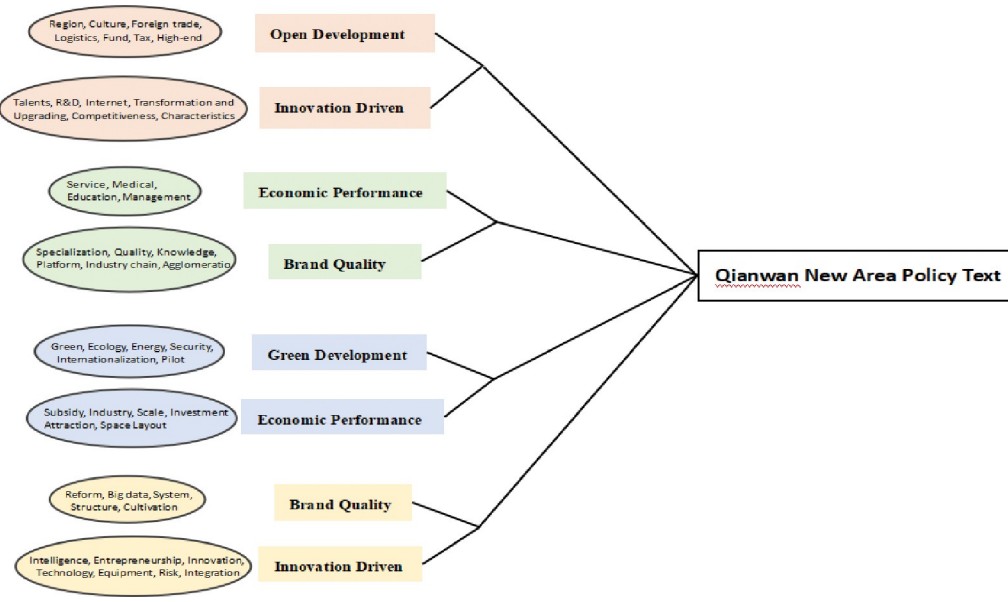

**Fig 6. Classification of keyword agglomeration subgroups in Qianwan New Area.**

manufacturing industry for the supply of professionals, and emphasizes the integration of industry and education and the development of multi industry agglomeration; The third category mainly reflects the two dimensions of economic benefits and green development, represented by key words such as scale, spatial layout, modernization, internationalization, quality, brand and green, reflecting the policy support for the open and large-scale green development of manufacturing industry in the bay area; The fourth category is reflected in the two dimensions of innovation driven and brand quality. Represented by keywords such as intelligence, technology, entrepreneurship, energy, reform, big data and cultivation, it reflects that the bay area manufacturing industry attaches importance to intelligent manufacturing and has a clear demand for scientific and technological innovation and talents. The stable clustering characteristics of this category of keywords and small differences in subgroup characteristics show that QianwanNew Area is committed to actively promoting innovative, digital, green and other characteristic industrial clusters, adhering to cultivating technical talents and helping the sustainable development of manufacturing industry in the bay area.

## 4.2 Policy network analysis of Guangdong-Hong Kong-Macao Bay Area

According to the department cooperation frequency of Guangdong, Hong Kong and Macao, the department cooperation matrix shown in Table 5 is obtained. Gephi is used to draw the department cooperation network diagram of Guangdong-Hong Kong-Macao Bay Area from 2009 to 2021, as shown in Fig 7, with a total of 19 participating departments. It can be seen from the cooperation network diagram of government departments that the Guangdong provincial government, the provincial development and Reform Commission and the Provincial Department of human resources and social security are at the core of the network, and cooperate frequently with other departments with a high degree of closeness, indicating that they have played an important role in the process of regional manufacturing policy-making. On the whole, there is a relatively stable intergovernmental cooperation relationship between the

**Table 5. Department cooperation matrix of Guangdong-Hong Kong-Macao Bay Area.**

| Department | A | B | C | D | E | F | G | H | I | J | K | L | M | N | O | P | Q | R | S |
|---|---|---|---|---|---|---|---|---|---|---|---|---|---|---|---|---|---|---|---|
| A | **0** | 4 | 4 | 2 | 2 | 1 | 1 | 2 | 2 | 1 | 4 | 1 | 1 | 1 | 3 | 3 | 1 | 1 | 1 |
| B | 4 | **0** | 1 | 2 | 2 | 1 | 1 | 2 | 3 | 1 | 2 | 1 | 1 | 1 | 2 | 2 | 1 | 1 | 1 |
| C | 4 | 1 | **0** | 1 | 1 | 1 | 1 | 1 | 1 | 0 | 0 | 0 | 0 | 0 | 1 | 0 | 0 | 0 | 0 |
| D | 2 | 2 | 1 | **0** | 2 | 1 | 1 | 2 | 2 | 0 | 1 | 0 | 0 | 0 | 2 | 1 | 0 | 0 | 1 |
| E | 2 | 2 | 1 | 2 | **0** | 1 | 1 | 2 | 2 | 0 | 1 | 0 | 0 | 0 | 2 | 1 | 0 | 0 | 1 |
| F | 1 | 1 | 1 | 1 | 1 | **0** | 1 | 2 | 2 | 1 | 1 | 1 | 1 | 1 | 2 | 1 | 1 | 1 | 0 |
| G | 1 | 1 | 1 | 1 | 1 | 1 | **0** | 1 | 1 | 0 | 0 | 0 | 0 | 0 | 1 | 0 | 0 | 0 | 0 |
| H | 2 | 2 | 1 | 2 | 2 | 2 | 1 | **0** | 3 | 1 | 2 | 1 | 1 | 1 | 3 | 2 | 1 | 1 | 1 |
| I | 2 | 3 | 1 | 2 | 2 | 2 | 1 | 3 | **0** | 1 | 2 | 1 | 1 | 1 | 3 | 2 | 1 | 1 | 1 |
| J | 1 | 1 | 0 | 0 | 0 | 1 | 0 | 1 | 1 | **0** | 1 | 1 | 1 | 1 | 1 | 1 | 1 | 1 | 0 |
| K | 4 | 2 | 0 | 1 | 1 | 1 | 0 | 2 | 2 | 1 | **0** | 1 | 1 | 1 | 2 | 2 | 1 | 1 | 1 |
| L | 1 | 1 | 0 | 0 | 0 | 1 | 0 | 1 | 1 | 1 | 1 | **0** | 1 | 1 | 1 | 1 | 1 | 1 | 0 |
| M | 1 | 1 | 0 | 0 | 0 | 1 | 0 | 1 | 1 | 1 | 1 | 1 | **0** | 1 | 1 | 1 | 1 | 1 | 0 |
| N | 1 | 1 | 0 | 0 | 0 | 1 | 0 | 1 | 1 | 1 | 1 | 1 | 1 | **0** | 1 | 1 | 1 | 1 | 0 |
| O | 3 | 2 | 1 | 2 | 2 | 2 | 1 | 3 | 3 | 1 | 2 | 1 | 1 | 1 | **0** | 2 | 1 | 1 | 1 |
| P | 3 | 2 | 0 | 1 | 1 | 1 | 0 | 2 | 2 | 1 | 2 | 1 | 1 | 1 | 2 | **0** | 1 | 1 | 1 |
| Q | 1 | 1 | 0 | 0 | 0 | 1 | 0 | 1 | 1 | 1 | 1 | 1 | 1 | 1 | 1 | 1 | **0** | 1 | 0 |
| R | 1 | 1 | 0 | 0 | 0 | 1 | 0 | 1 | 1 | 1 | 1 | 1 | 1 | 1 | 1 | 1 | 1 | **0** | 0 |
| S | 1 | 1 | 0 | 1 | 1 | 0 | 0 | 1 | 1 | 0 | 1 | 0 | 0 | 0 | 1 | 1 | 0 | 0 | **0** |

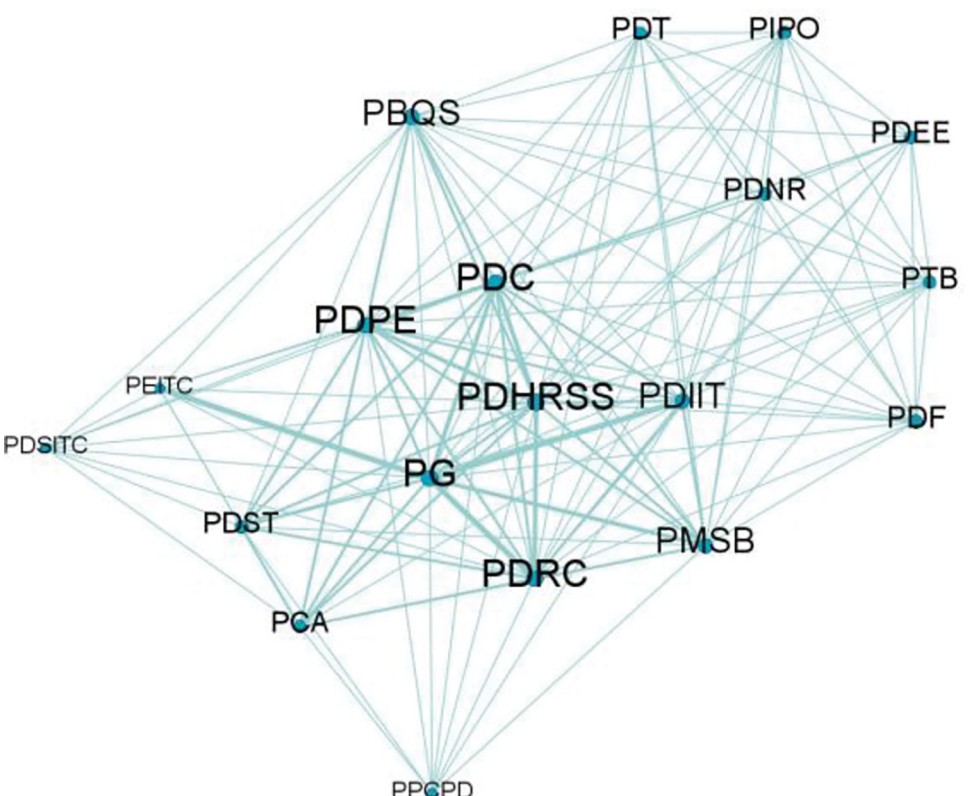

**Fig 7. Department cooperation network of Guangdong-Hong Kong-Macao Bay Area.**

government departments of Guangdong, Hong Kong and Macao.(Please refer to S1 Appendix at the end of the text for the names of government departments represented by the letters in the matrix).

**4.2.1 Analysis of cooperation between policy participating departments.** *(1) Centrality analysis*. Use UCINET6.0 and SPSS to draw the Department centrality map of Guangdong-Hong Kong-Macao Great Bay Area as shown in the figure, as shown in Fig 8. With the mean value of point centrality as the horizontal axis and the mean value of intermediate centrality as the vertical axis, it is divided into four quadrants. The number label in the figure corresponds to the serial number of government departments in Table 4. It is concluded from the figure that the five departments of Guangdong Provincial People's government, Provincial Department of human resources and social security, Provincial Department of Commerce, Provincial Department of education and provincial development and Reform Commission are all in a higher position in the first quadrant. It shows that the people's Government of Guangdong Province has carried out decision-making research on the key points of regional economic development, leading the direction for the sustainable development of manufacturing industry in the bay area; The Provincial Department of human resources and social security and the Provincial Department of education are responsible for formulating policies related to talent introduction, employment and entrepreneurship, intelligent transformation and upgrading, and coordinating and solving the problem of talent supply for the manufacturing industry in the bay area; The Provincial Department of Commerce strongly supports the manufacturing industry of processing trade and promotes the sustainable development of trade in the bay area; The provincial development and Reform Commission is responsible for defining regional

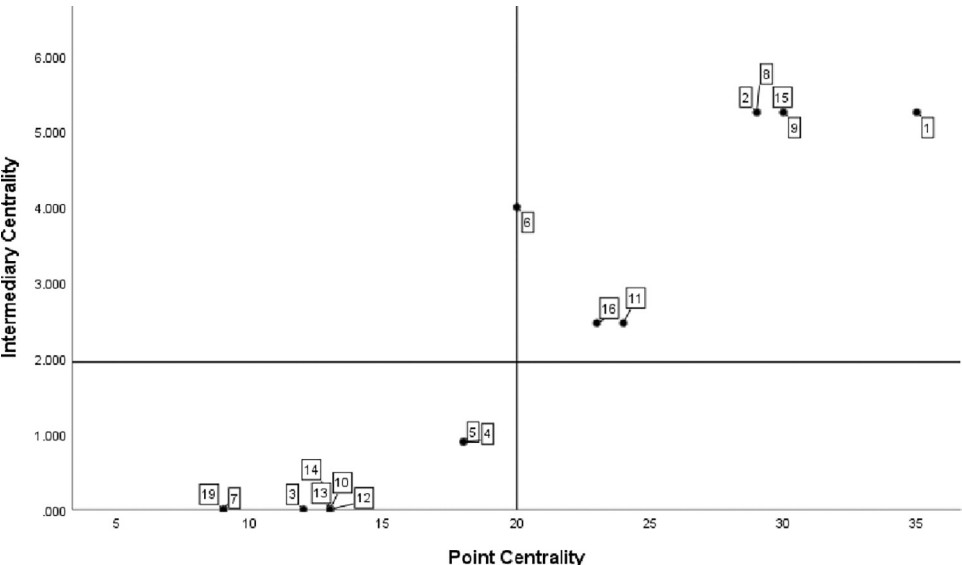

**Fig 8. Analysis of centrality of government departments in Guangdong-Hong Kong-Macao Bay Area.**

priority industries and providing development objectives and benefit policies for manufacturing enterprises. The network information office of the provincial Party committee and the Publicity Department of the provincial Party committee are in the third quadrant. The reason is that the main work of the network information office of the provincial Party committee focuses on Internet related work. The Publicity Department of the provincial Party committee focuses on publicizing the policy spirit of government departments. They have relatively little contact with other manufacturing related departments and are in a marginal position in the central network.

*(2) Structural hole analysis*. Use UCINET6.0 to obtain the hole index of department structure, as shown in Table 6. Combining the two indicators, the Guangdong provincial government, the Provincial Department of education, the Provincial Department of human resources and social security and the Provincial Department of Commerce have a larger effective scale and less restrictions in the network. It shows that the above four departments have played an important role in the resource allocation of manufacturing industry in the bay area, especially making full use of the dominant position of the provincial government in resource

**Table 6. Structural indicators of government departments in Guangdong-Hong Kong-Macao Bay Area.**

| Serial number | Department | Effective scale | Limit degree | Serial number | Department | Effective scale | Limit degree |
|---|---|---|---|---|---|---|---|
| 1 | PG | **8.105** | **0.231** | 11 | PDIIT | 5.285 | 0.27 |
| 2 | PDRC | 7.351 | 0.245 | 12 | PDF | 4.404 | 0.294 |
| 3 | PEITC | 3.819 | 0.353 | 13 | PDEE | 4.404 | 0.294 |
| 4 | PDST | 4.963 | 0.337 | 14 | PDT | 4.404 | 0.294 |
| 5 | PCA | 4.963 | 0.337 | 15 | PDC | **7.578** | **0.24** |
| 6 | PBQS | 7.317 | 0.243 | 16 | PMSB | 5.493 | 0.265 |
| 7 | PNSITC | 3.537 | 0.422 | 17 | PTB | 4.404 | 0.294 |
| 8 | PDPE | **7.716** | **0.238** | 18 | PIPO | 4.404 | 0.294 |
| 9 | PDHRSS | **7.631** | **0.239** | 19 | PPCPD | 3.231 | 0.423 |
| 10 | PDNR | 4.404 | 0.294 | | | | |

coordination to coordinate the participation of other departments in policy-making. The Provincial Department of education is in the second important position, which is related to the implementation of the integration of industry and education and the response to the digital and intelligent transformation and upgrading of the manufacturing industry. As the auxiliary departments for the sustainable development of manufacturing industry in the bay area, the network information office of Guangdong provincial Party committee and the Publicity Department of Guangdong provincial Party committee have small effective scale and great restrictions. Therefore, they have low relevance to the development of manufacturing policies and weak ability to coordinate resources.(Please refer to S1 Appendix at the end of the text for the names of government departments represented by the letters in the matrix).

**4.2.2 Co-occurrence analysis of policy keywords.** Import keyword matrix into UCI-NET6.0 Use Netdraw to draw the keyword co-occurrence network diagram of Guangdong, Hong Kong and Macao Bay Area, which can intuitively analyze the relationship between keywords, as shown in Fig 9.

From the map of keyword co-occurrence network, we can see that manufacturing keywords have different degrees of distribution. Keywords such as cluster, talent, technology, innovation, equipment and service are in the core position in the co-occurrence network, followed by capital, environment, platform and industrial chain. In order to promote the development of advanced manufacturing clusters, Guangdong-Hong Kong-Macao Dawan district has selected

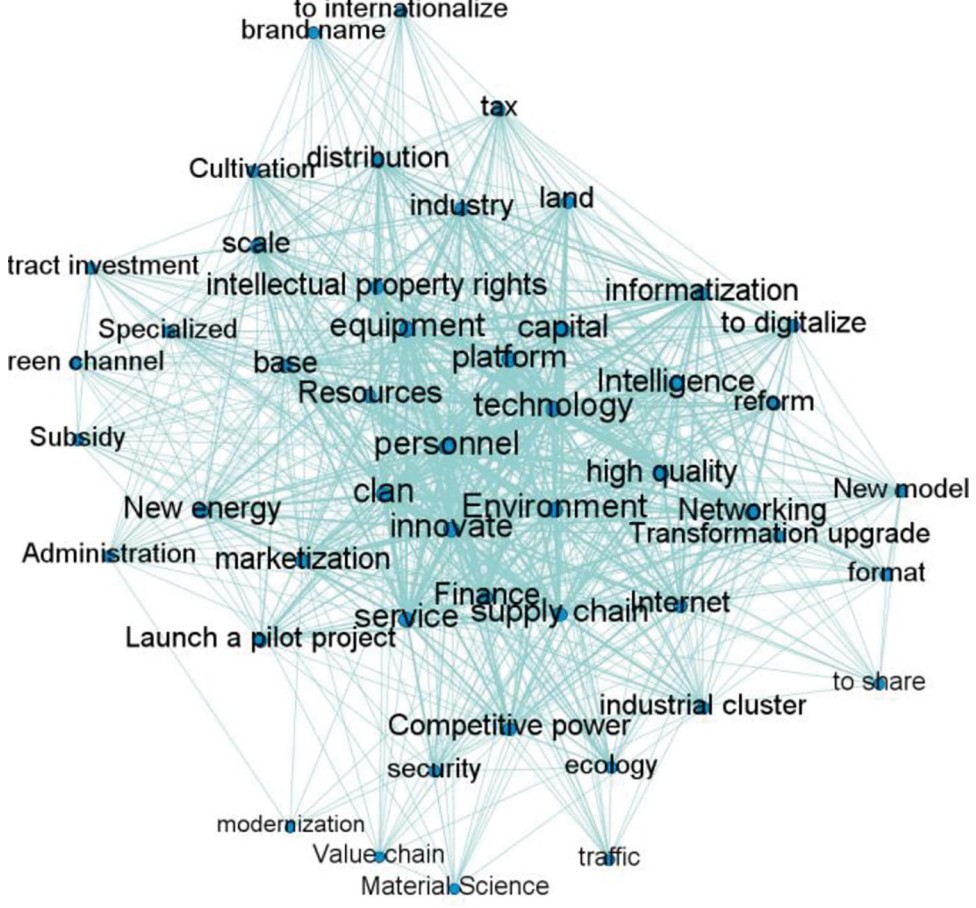

**Fig 9. Keyword co-occurrence network of Guangdong-Hong Kong-Macao Bay Area.**

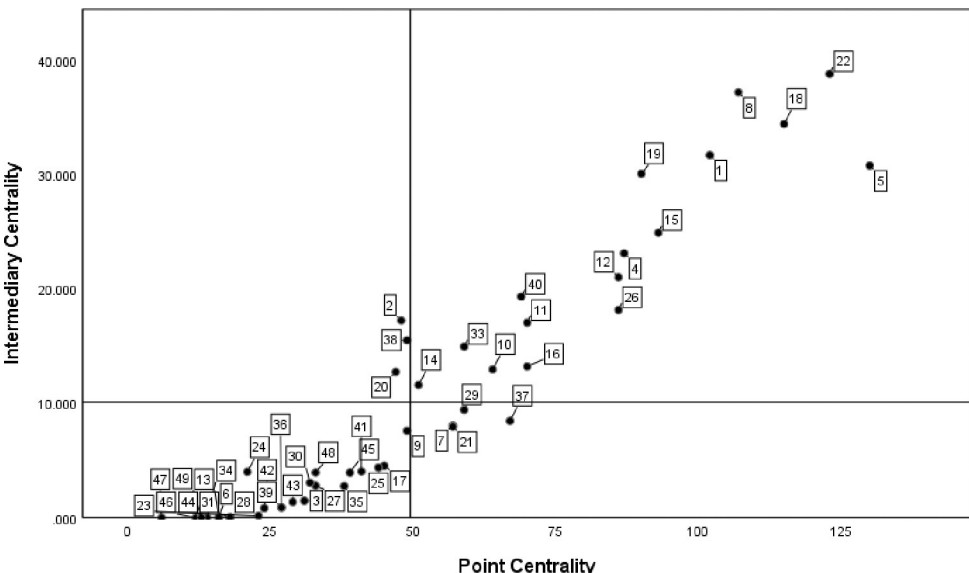

**Fig 10. Value chart of policy keyword point centrality and intermediary centrality.**

key areas such as new generation information technology, high-end equipment and manufacturing services as key cultivation objects, and focused on increasing policy support for talents, financial funds and so on.

*(1) Centrality analysis.* The scatter diagram of policy keyword centrality is shown in Fig 10. Talents, innovation, technology, cluster, equipment, service and other keywords are in the first quadrant, indicating that the keyword group is in the core position in the co-occurrence network and occupies a leading position in the policy text. It is the main focus of the bay area's sustainable development support policy for manufacturing industry. At the same time, it also shows that Guangdong-Hong Kong-Macao Bay Area is committed to building an industrial group with the characteristics of talents, technology, innovation and equipment manufacturing industry. In addition, it can be found that most of the other keywords are located in the third quadrant, such as specialization, finance, transportation, green channel, value chain and other keywords. The intermediary centrality and point centrality are low, indicating that such policy keywords only appear in a few policy texts, the support for policy texts is relatively weak, and the impact on other keywords is also small.

*(2) Structural hole analysis.* Use UCINET6.0 to calculate the structural hole index of policy keywords, and the results are shown in Table 7. Equipment, technology, cluster, innovation and talents have the highest effective scale and the lowest restriction, indicating that these five keywords are more likely to have structural holes in the network and more likely to appear in the manufacturing policy text; The restrictions of modernization, finance, value chain and materials are higher than 0.3, among which "modernization" is more than 0.6. The high limit system of such keywords shows that it has weak power to obtain redundant information through structural holes in the network and its intermediary role in the keyword network. At the same time, it also shows that it is vulnerable to the change of policy subjects and is scattered in the manufacturing policy text.

*(3) Condensed subgroup analysis.* According to the network characteristics of policy keywords, the manufacturing policy keywords of Guangdong, Hong Kong and Macao Dawan district are analyzed by cohesive subgroup to reveal the deep relationship between keywords and subgroups, as shown in the subgroup classification diagram in Fig 11.

**Table 7. Keyword structure hole index table of manufacturing policy in Guangdong-Hong Kong-Macao Bay Area.**

| Serial number | Keyword | Effective scale | Limit degree | Serial number | Keyword | Effective scale | Limit degree | Serial number | Keyword | Effective scale | Limit degree |
|---|---|---|---|---|---|---|---|---|---|---|---|
| 1 | Equipment | 26.501 | **0.111** | 18 | Innovate | 26.531 | **0.113** | 35 | Reform | 12.859 | 0.173 |
| 2 | Marketization | 18.947 | 0.138 | 19 | Service | 25.714 | 0.116 | 36 | Business type | 11.055 | 0.188 |
| 3 | Cultivate | 11.28 | 0.184 | 20 | New energy | 18.558 | 0.14 | 37 | Networking | 18.705 | 0.135 |
| 4 | Capital | 23.36 | 0.12 | 21 | Resource | 18.679 | 0.136 | 38 | Competitive power | 17.266 | 0.155 |
| 5 | Technology | 26.461 | **0.11** | 22 | Personnel | 26.633 | **0.111** | 39 | New mode | 11.055 | 0.188 |
| 6 | Brand | 7.117 | 0.247 | 23 | Specialization | 8.78 | 0.222 | 40 | High quality | 20.59 | 0.135 |
| 7 | Industry | 16.97 | 0.146 | 24 | Administration | 10.943 | 0.207 | 41 | Digitization | 16.532 | 0.147 |
| 8 | Colony | 27.431 | **0.111** | 25 | Land | 15.356 | 0.151 | 42 | Security | 9.246 | 0.212 |
| 9 | Scale | 15.856 | 0.148 | 26 | Industrial chain | 21.771 | 0.124 | 43 | Ecology | 9.95 | 0.207 |
| 10 | Base | 19.348 | 0.133 | 27 | Experimental unit | 12.118 | 0.175 | 44 | Traffic | 6.026 | 0.285 |
| 11 | Intelligent | 21.94 | 0.124 | 28 | Subsidy | 7.777 | 0.222 | 45 | Transformation and Upgrading | 13.893 | 0.163 |
| 12 | Platform | 23.104 | 0.121 | 29 | Finance | 17.664 | 0.143 | 46 | Finance | 5.086 | 0.329 |
| 13 | Internationalization | 7.117 | 0.247 | 30 | Tax revenue | 13.793 | 0.16 | 47 | Modernization | 2.986 | 0.624 |
| 14 | Intellectual property right | 18.621 | 0.136 | 31 | Green channel | 7.777 | 0.222 | 48 | Industrial Cluster | 14.533 | 0.16 |
| 15 | Environment | 23.959 | 0.119 | 32 | Attracting investment | 7.777 | 0.222 | 49 | Value chain | 5.801 | 0.302 |
| 16 | Informatization | 20.856 | 0.129 | 33 | Internet | 18.548 | 0.141 | 50 | Material | 5.801 | 0.302 |
| 17 | Logistics | 15.416 | 0.153 | 34 | Sharing | 6.23 | 0.283 | | | | |

As can be seen from the figure, the policy keywords of Guangdong-Hong Kong-Macao Bay Area can be divided into 8 subgroups and grouped into 4 categories. Although the number of subgroups is the same as that of QianwanNew Area, it is obvious that the density difference between subgroups in Guangdong, Hong Kong and Macao is large. The first category, represented by capital, finance, technology, informatization, brand, internationalization, logistics

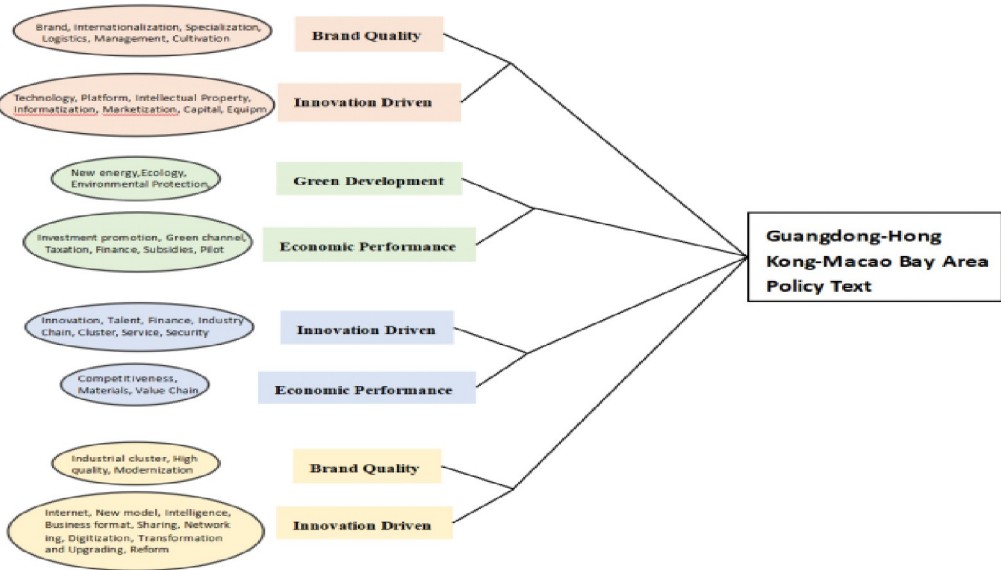

**Fig 11. Classification of keyword agglomeration subgroups in the Guangdong-Hong Kong-Macao Bay Area.**

and management, reflects the support of Guangdong, Hong Kong and Macao Dawan district for manufacturing capital, technology management and open development, including two subgroups; The second category is represented by scale, base, new energy, taxation, subsidies, investment attraction and pilot projects, reflecting the requirements for supporting the industrial factor resources of manufacturing industry in Dawan District, including two subgroups; The third category, represented by clusters, ecology, talents, finance, competitiveness, value chain and materials, reflects and supports the reconstruction of regional value chain and realizes the diversified integration of ecology, capital and talents, including two subgroups; The fourth category, represented by sustainability, modernization, industrial clusters, intelligence, Internet, digitization, reform and sharing, reflects that Dawan district is committed to a new industrial revolution and gives full play to the role of Internet, big data and artificial intelligence, including two subgroups. The study found that the number of subgroups divided by keywords is large, which to a certain extent shows that the policy subject can give comprehensive policy support to the manufacturing industry by considering the industrial development direction and development hot spots in a diversified way when formulating policies.

## 5. Comparative analysis of the characteristics of the policy network in the two bay areas

### 5.1 Close cooperation between departments

The group eigenvalues of government departments and policy text keyword networks in the two bay areas are calculated by social network analysis method. The Group Centrality and group intermediary centrality are calculated by Eq (1) and Eq (2) respectively, and the comparison of the cooperation tightness of the two systems is obtained, as shown in Table 8.

**(1) Close cooperation between departments.** The Department cluster centrality of QianwanNew Area is greater than that of Guangdong, Hong Kong and Macao Dawan District, which reflects that there is a greater possibility of a particularly important core department in its department network. According to the previous data analysis, the Management Committee of Ningbo Qianwan New Area plays a leading role in the development of regional manufacturing industry, and other departments assist it in the formulation and implementation of manufacturing policies; In terms of department network density and department group intermediary degree, QianwanNew Area is smaller than Guangdong, Hong Kong and Macao Dawan District, which reflects that Guangdong, Hong Kong and Macao Dawan district has more department cooperation behavior, greater mutual support and better information circulation, which helps to improve the overall work performance of the Department. At the same time, there is one or more most critical or important departments, which have a large number of close cooperation and exchanges with other departments and high leadership in the Department network.

**(2) Keywords compactness.** The keyword group intermediation degree of QianwanNew Area is greater than that of Guangdong, Hong Kong and Macao Dawan District, while its

**Table 8. Comparison of overall cooperation tightness between the two bay areas.**

| Index | Departmental network density | Department Group Centrality | Intermediary degree of department group | Keyword network density | Keyword Group Centrality | Keyword group intermediation |
|---|---|---|---|---|---|---|
| Qianwan New Area | 0.7111 | 0.3889 | 0.1883 | 0.7264 | 0.2367 | 0.029 |
| Guangdong-Hong Kong-Macao Bay Area | 0.784 | 0.2459 | 0.0228 | 0.588 | 0.2442 | 0.0249 |

keyword network density and keyword Group Centrality are less than that of Guangdong, Hong Kong and Macao Dawan District, which reflects that compared with Guangdong, Hong Kong and Macao Dawan District, the manufacturing policy text of QianwanNew Area has a low contribution rate to the realization of the manufacturing policy objectives of the Bay District, but its high keyword group intermediation degree indicates that the focus of the policy text is more concentrated, mainly reflected in industrial clusters, technological innovation Talent supply and other aspects.

## 5.2 Comparative analysis of keyword network characteristics

In this paper, python algorithm is used to extract keywords from the policy texts of Qianwan-New Area and Guangdong-Hong Kong-Macao Great Bay Area. The keyword frequency is counted from high to low, and the top 10 keywords with the highest frequency are selected respectively, as shown in Table 9.

Combined with the policy text analysis, QianwanNew Area pays more attention to intelligent manufacturing, integration with service industry, cultivation of management talents, government subsidies and other aspects, which reflect the hot spots and priorities of the current manufacturing development of the Bay area, as well as the objectives and requirements for future development. However, compared with Guangdong, Hong Kong and Macao Dawan District, Qianwan New District has less frequency of high-frequency words and smaller frequency gap; The high-frequency keywords of Guangdong-Hong Kong-Macao Bay Area mainly focus on innovation, technology, service, cluster, Internet and platform, indicating that Guangdong-Hong Kong-Macao Bay Area attaches great importance to the field of manufacturing technological innovation. At the same time, the integration of service industry and industrial cluster are also the focus of the government.

## 5.3 Comparative analysis of department keyword relevance

In order to further analyze the subordinate relationship between policy subjects and objectives, this paper uses UCINET6.0 The membership relationship of departments and keywords in the two bay areas is analyzed by singular value decomposition (SVD) method in 0. According to the Department keyword affiliation of the two Bay districts, the proportion of government departments in the Bay District participating in policy-making types is analyzed. According to the proportion of government departments participating in keywords, the top five departments in the two Bay districts are selected from large to small, and the proportion table of department participating in key words is compiled as shown in Table 10. (Please refer to S1 Appendix at the end of the text for the names of government departments represented by the letters in the matrix).

Firstly, from the perspective of the proportion of departments corresponding to coordination keywords, the government departments of the two Bay districts have correspondingly

**Table 9. Frequency of key words in policy texts of two bay areas.**

| | ① | ② | ③ | ④ | ⑤ | ⑥ | ⑦ | ⑧ | ⑨ | ⑩ |
|---|---|---|---|---|---|---|---|---|---|---|
| Qianwan New Area | Intelligence | Service | Cultivation | Subsidy | Quality | Administration | System | Technology | Personnel | Platform |
| Frequency | 67 | 64 | 42 | 41 | 38 | 33 | 29 | 29 | 27 | 23 |
| Guangdong-Hong Kong-Macao Bay Area | Innovate | Technology | Service | Colony | Internet | Platform | Equipment | Base | Material | Digitization |
| Frequency | 438 | 364 | 312 | 254 | 244 | 213 | 196 | 185 | 166 | 157 |

**Table 10. Percentage of keywords involved in the two major may area departments.**

| Bay Area | Department | Number of Keywords | The Percentage of Keywords | Bay Area | Department | Number of Keywords | The Percentage of Keywords |
|---|---|---|---|---|---|---|---|
| Ningbo Qianwan New Area | DMC | 118 | 35.65% | Guangdong-Hong Kong-Macao BayArea | PG | 153 | 23.29% |
| | DEDB | 45 | 13.60% | | PEITC | 62 | 9.44% |
| | DMSA | 34 | 10.27% | | PDIIT | 59 | 8.98% |
| | DCB | 33 | 9.97% | | PDRC | 47 | 7.15% |
| | DHRSSB | 21 | 6.34% | | PDHRSS | 45 | 6.85% |

participated in the manufacturing policy-making process related to them on the basis of their own departmental functions.

Secondly, the average number of keywords involved in relevant policies of each department is 33, and the average number of departments involved in each keyword is 6.6. Moreover, the subordinate relationship between departments and keywords is sparse, and the corresponding participation of departments varies greatly. The number of keywords involved by the Management Committee of Qianwan New Area accounts for the largest proportion. In the manufacturing policy keyword group, the main participating keywords are subsidies, services, medical treatment, R & D, management, transformation and upgrading, etc; The average number of relevant policy keywords participated by each department in Guangdong, Hong Kong and Macao Dawan district is 35, and the average number of participating departments of each keyword is 14.8. The Department keyword subordinate relationship is close, and the corresponding keyword leadership of each department has little difference. The Guangdong Provincial People's government participates in the largest number of keywords. In the manufacturing policy keyword group, the main participating keywords are equipment, technology, cluster, talent, innovation, service, etc.

Finally, by comparing the Department keyword correlation degree of Guangdong, Hong Kong and Macao Dawan District, it can be found that there is little difference in the average number of keywords involved in each department between the two Dawan districts, indicating that the government departments of the two Dawan districts have the same degree of investment in the process of manufacturing policy participation. However, the average number of participating departments of each keyword in QianwanNew Area is significantly less than that in Guangdong, Hong Kong and Macao Dawan District, indicating that all units need to improve the focus of policies from their own departmental functions.

## 6. Evaluation of high quality development system of manufacturing industry in bay area

### 6.1 Index selection

In order to scientifically and comprehensively evaluate the sustainable development quality of manufacturing industry, this paper comprehensively refers to the sustainable development index system of manufacturing industry proposed by scholars [55–57], takes into account the principles of comprehensiveness, representativeness and operability, and finally selects five dimensions of innovation driven, economic benefits, brand quality, green development and open development as research indicators, Based on the measurement method of policy synergy degree proposed by Tang H et al. [58], through the network density, Group Centrality and group intermediary centrality in the departmental cooperation network and keyword co-occurrence network, this paper builds the sustainable development index system of manufacturing industry in QianwanNew Area and Guangdong, Hong Kong and Macao Bay

Area, and quantitatively evaluate the relationship between manufacturing policy supply and sustainable development in the bay area.

## 6.2 Construction of evaluation system

The sustainable development of manufacturing industry in the two bay areas is measured from five dimensions: innovation driven, economic benefits, brand quality, green development and open development. The index calculation is shown in Table 11. In the process of selecting indicators, according to the principle of representativeness and operability, the indicators corresponding to the dimension of sustainable development and easy to quantify are selected, and each dimension corresponds to two indicators.

Select the data calculated by the sustainable evaluation index system in Table 11, and use the range method (Eq (3)) to carry out dimensionless processing, and obtain the standardized value after conversion. In order to avoid the influence of value 0 on the calculation of sustainable index, intercept processing (0.01) is carried out on the mark value during the conversion process. At the same time, since the energy consumption per unit industrial value (X7) in Table 10 is a reverse index, it needs to be calculated by taking the reciprocal. Considering that the information content and significance of the 10 indicators in sustainable development are different, and their weights are also different, it is necessary to calculate the weight of each indicator through the discrete coefficient method (Eq (4)), and calculate the mean Xi and standard deviation of each indicator δ I calculate and divide to obtain the discrete coefficient, then determine the index weight according to the proportion of the discrete coefficient, and finally summarize the value of each sustainable development index corresponding to the two bay areas.

## 6.3 Comparative analysis of sustainable development system in two bay areas

The comprehensive indexes of five sustainable development dimensions of two bay areas are calculated, as shown in Table 12. From the comparative analysis of the five dimensions in the table, it can be seen that QianwanNew Area has more advantages than Guangdong, Hong Kong and Macao Dawan area in the process of innovation driven, economic benefit and brand quality development, indicating that it has a relatively sound innovation system, efficient

**Table 11. Calculation of high quality evaluation index.**

| High Quality Dimension | Index | Computing Method |
|---|---|---|
| Innovation Driven (D1) | Innovation investment (X1) | R & D expenditure / main business income |
| | Innovation level (X2) | New product sales revenue / main business revenue |
| Economic Benefits (D2) | Labor productivity (X3) | Industrial added value / average number of employees |
| | Return on net assets (X4) | Total operating profit / main business income |
| Brand Quality (D3) | Product quality qualification rate (X5) | Number of qualified products / finished products of manufacturing enterprises |
| | Invention patent (X6) | Number of effective invention patents of Industrial Enterprises above Designated Size |
| Green Development (D4) | Energy consumption per unit industrial value (X7) | Total energy consumption / total industrial output value |
| | Comprehensive utilization rate of general industrial solid waste (X8) | Comprehensive utilization of industrial solid waste / production of industrial solid waste |
| Open Development (D5) | Foreign trade openness (X9) | Total import and export / total industrial output value |
| | Attracting foreign investment (X10) | Foreign direct investment / gross industrial output value |

**Table 12. Bay area manufacturing sustainable development composite index table.**

| Index | Qianwan New Area | Guangdong-Hong Kong-Macao Bay Area |
|---|---|---|
| Innovation Driven | 1.073 | 0.947 |
| Economic Performance | 0.811 | 0.296 |
| Brand Quality | 1.513 | 0.923 |
| Green Development | 0.224 | 1.453 |
| Open Development | 0.545 | 1.635 |

economic system and good brand image. Guangdong, Hong Kong and Macao's Great Bay Area has performed significantly better in the process of green development and open development, which is related to the central and local attaching great importance to its green competitiveness and the development level of international competitiveness.

Select the sustainable development indicators of the two bay areas from 2011 to 2020, and draw the trend chart of the sustainable development indicators of the manufacturing industry in the bay area respectively, as shown in Figs 12 and 13, so as to show the differences between the sustainable development of the manufacturing industry in the two bay areas.

The following conclusions can be drawn by comparing the five dimensions in the figure:

**(1) Innovation driven.** On the whole, the indexes of the two bay areas showed an upward trend. Thanks to the loose scientific and technological innovation environment in Zhejiang Province, although the rise of QianwanNew Area is relatively flat, its income generation started earlier than that of Guangdong, Hong Kong and Macao Dawan district. It can be seen that QianwanNew Area has a relatively complete innovation system in promoting the sustainable development of manufacturing industry. There is a turning point in the development trend of Guangdong-Hong Kong-Macao Bay Area. After 2015, the speed of innovation and development in this area has significantly accelerated, reflecting that the development of technological innovation capital in Guangdong-Hong Kong-Macao Bay Area is more mature, and the capital support of Hong Kong and Macao makes the innovation resources in this area richer.

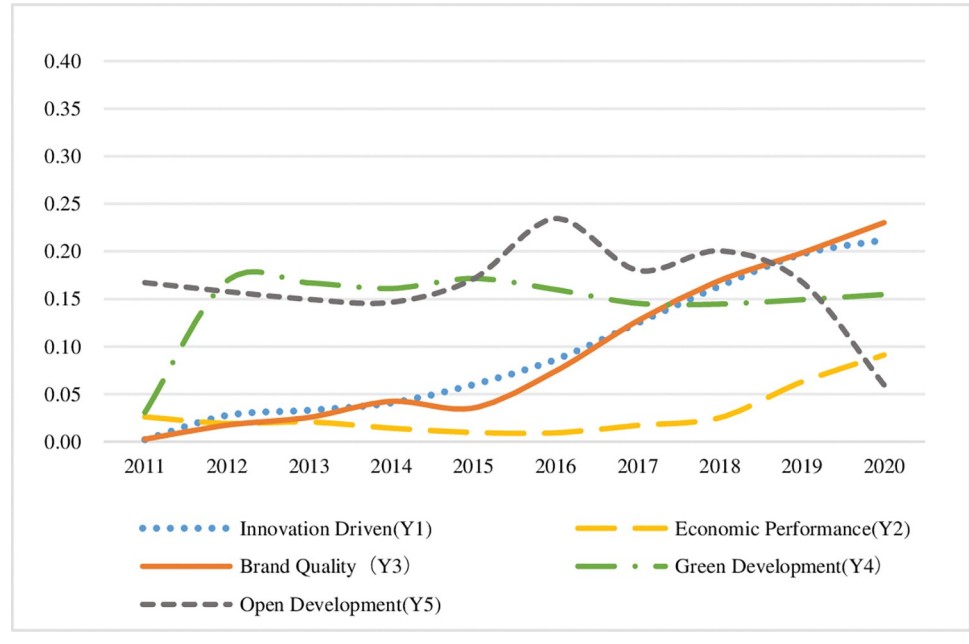

**Fig 12. Trends of sustainable development indicators in QianwanNew Area.**

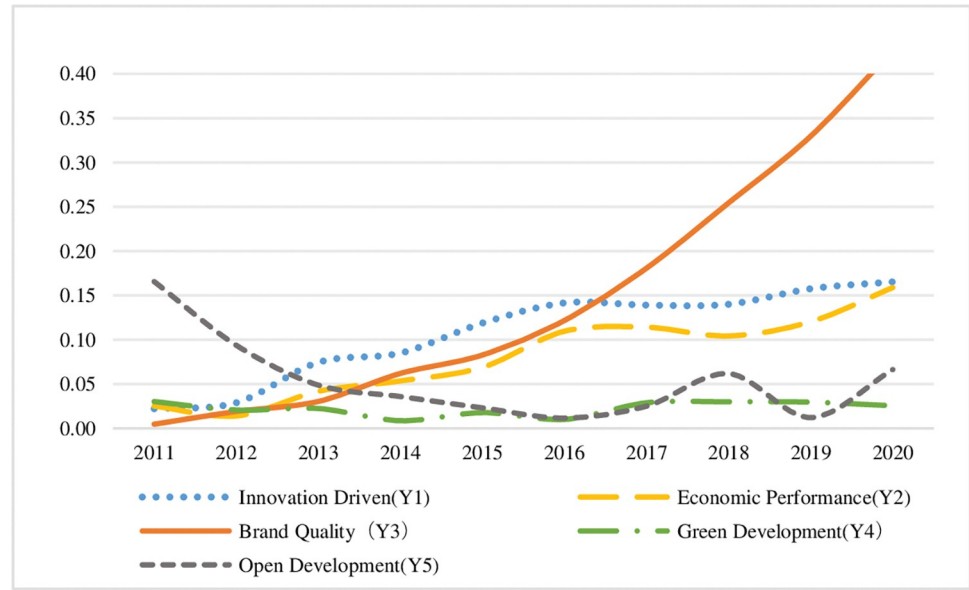

**Fig 13. Trends of sustainable development indicators in the Guangdong-Hong Kong-Macao Bay Area.**

**(2) Economic benefits.** On the whole, the indexes of the two bay areas show an upward trend, but the rising range of QianwanNew Area is greater than that of Guangdong, Hong Kong and Macao Bay Area, indicating that QianwanNew Area has obvious advantages in manufacturing economic competition relying on regional advantages. The Guangdong-Hong Kong-Macao Bay Area was in a long-term low development state from 2011 to 2017, with low economic benefits. However, the economic benefits of manufacturing industry have improved since 2018, thanks to the gradual rise of Guangdong-Hong Kong-Macao Bay Area into a national strategy after 2017, and the development of manufacturing industry has been supported by the policies of the central and local governments.

**(3) Brand quality.** The overall indexes of the two bay areas show a significant growth trend. In particular, QianwanNew Area has maintained a high-speed growth trend year after year, and gradually opened the gap with other dimensions. Among the five dimensions, it has made the greatest contribution to the sustainable development of manufacturing industry, indicating that the region has always attached great importance to the construction of manufacturing brand image and innovation, R & D and invention patents. Although the overall increase of Guangdong-Hong Kong-Macao Bay Area index is less than that of Qianwan-New Area, the development of manufacturing brand quality in the bay area began to enter a state after 2015, which is related to the dislocation development characteristics of manufacturing development after the bay area introduced a large number of talents from surrounding urban agglomerations.

**(4) Green development.** There are obvious differences in the development trend of the index between the two bay areas. QianwanNew Area has basically been in a state of low development in the past 10 years, and has made the smallest contribution to the sustainable development of manufacturing industry in the five dimensions. The reason is that the energy resource utilization efficiency of manufacturing industry in the bay area is not high, and the government and enterprises need to cooperate to adjust the energy consumption structure. In the past decade, Guangdong, Hong Kong and Macao Dawan district has deeply implemented the concept of green development and fully implemented the green development of

manufacturing industry. Since 2011, the quality of green development in the Bay District has been significantly improved, and then its index has maintained a relatively stable high development trend. It can be seen that the green competitiveness level of manufacturing industry in Guangdong, Hong Kong and Macao Dawan district is relatively good.

**(5) Open development.** The indexes of the two bay areas showed a downward trend of fluctuation as a whole, and there was a trough in the open development index during this period, but there was a significant difference in the time point of significant decline. The opening-up development index of QianwanNew Area decreased significantly from 2011 to 2016. The reason is that during this period, the foreign trade enterprises in the bay area encountered problems such as market shrinkage, rising costs and financing difficulties. Later, with the policy support of cross-border e-commerce and promoting the development of foreign trade, the opening-up development situation improved. In 2020, the opening-up development index of Guangdong, Hong Kong and Macao Dawan District plummeted, the quality of industrial opening-up development decreased, and its contribution to sustainable development no longer has an advantage.

## 7. Conclusions and policy implications

This study mainly uses the social network analysis method to obtain the similarities and differences between the government departments of the two bay areas in terms of policy-making participation and policy text focus, and reveals the cooperation network, keyword network and sustainable development trend of manufacturing industry in the bay area represented by Ningbo Qianwan New Area and Guangdong, Hong Kong and Macao Bay Area. The main conclusions are as follows:(1)The degree of departmental cooperation plays a leading role in the policy effect. From the departmental network density, it can be seen that the cooperation between the government functional departments of the two bay areas is at a high level, and the core departments (Ningbo Qianwan New Area Management Committee and Guangdong Provincial People's Government) have the highest enthusiasm in the process of policy formulation and participation, which fully shows their core leadership in the economic development of manufacturing industry in the bay area. At the same time, the coordination ability of various departments in the process of resource allocation is the fundamental requirement for the realization of policy effect.(2)The keyword group cohesion is weak, and the policy objectives are scattered, which affects the policy implementation results. From the analysis of keyword group centrality, group intermediary centrality and cohesion subgroup, the keyword focus of the relevant policy texts of the manufacturing industry in the two bay areas needs to be further improved. Compared with Qianwan New Area, the policy keyword cohesion of Guangdong-Hong Kong-Macao Bay Area is stronger, and the development goal of manufacturing policy is more clear. Keyword network density will affect the allocation of resource elements and resource utilization of major industries, thus affecting the actual effect of policies. (3)There are significant spatial differences in the sustainable development of manufacturing industry in the bay area. The evaluation results of sustainable development indicators show that the two bay areas have excellent performance in the five sustainable development indicators, and the manufacturing industry in QianwanNew Area has relatively more advantages in the development process of innovation driven, economic benefits and brand quality. The Greater Bay Area of Guangdong, Hong Kong and Macao has strong policy support in the process of green development and open development. Generally speaking, the sustainable development of manufacturing industry in China's bay area is closely related to policy subjects, policy objectives and regional conditions.

Based on the conclusion, drawing on the advantages of the sustainable development of manufacturing industry in the two bay areas, this paper puts forward the following suggestions for the bay area government departments under the background of economic integration:(1) Build a high-density departmental cooperation network to improve the operation efficiency. While maintaining the high-level resource coordination ability of the core departments, strengthen the cooperation with other departments, coordinate the relationship between policy objectives, and effectively maximize the policy effect. (2)Identify regional advantages and empower the bay area according to local conditions. Through the keyword network, accurately locate the regional development advantages, focus on the key industrial fields of the bay area, encourage the mutual integration of industrial clusters and technology clusters, and give full play to the dual agglomeration advantages of the central area of the bay area. (3)Formulate differentiation policies to realize the complementary advantages of the Bay area. For low-level development indicators, learn from the general experience of other Bay areas, clarify the division of labor in the Bay area, and realize diversified and differentiated development according to factor endowment.

Relatively speaking, the qualitative analysis of sustainable development policy texts is more explanatory, inductive and exploratory, while the quantitative text analysis based on artificial intelligence is often more macro, deductive and objective. However, the analysis based on words will undoubtedly lead to the segmentation of certain fixed phrases or important collocation in statistics. In the interpretation, because of the inability to restore the context, the meaning of some words may be simplified or misunderstood. Through social network analysis, this paper reveals the policy framework for the sustainable development of the manufacturing industry in the Bay Area, and provides a more three-dimensional understanding of the formation mechanism of policy effects, which has strong explanatory depth and initiative advantages. However, there is still a lack of effective quantitative evaluation model for evaluating the effectiveness of sustainable economic development policies by using policy text network analysis.

## Supporting information

**S1 Appendix. List of covered pilot regions.**
(DOCX)

## Acknowledgments

We would like to show our greatest appreciation to anonymous reviewers, editor, County Industrial Digitization Research Base (the sixth round of Ningbo Philosophy and Social Science Research Base), Mei-Fang Zhang (Anhui University of Science and Technology) and those who have helped to contribute to this paper writing.

## Author Contributions

**Conceptualization:** Huijie Zhou.

**Data curation:** Huijie Zhou.

**Formal analysis:** Huijie Zhou.

**Funding acquisition:** Huijie Zhou.

**Investigation:** Huijie Zhou, Shangjia Yu.

**Project administration:** Shangjia Yu.

**Resources:** Shangjia Yu.

**Software:** Pengyue Wu.

**Supervision:** Pengyue Wu.

**Validation:** Pengyue Wu.

**Visualization:** Pengyue Wu.

**Writing – original draft:** Huijie Zhou.

**Writing – review & editing:** Huijie Zhou.

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
