## [Decision Letter · Decision Letter 0]

25 Sep 2023

PONE-D-23-23466Analyzing the impact of sustainable economic development from the policy text network: Based on the practice of China's Bay area policyPLOS ONE

Dear Dr. Zhou,

Thank you for submitting your manuscript to PLOS ONE. After careful consideration, we feel that it has merit but does not fully meet PLOS ONE’s publication criteria as it currently stands. Therefore, we invite you to submit a revised version of the manuscript that addresses the points raised during the review process.

We look forward to receiving your revised manuscript.

Kind regards,

Kashif Iqbal, Ph.D.,

Academic Editor

PLOS ONE

Journal Requirements:

"This work received support from Zhejiang Province Soft Science Research Project (2022C35101)."

"There are no financial conflicts of interest to disclose.This work received support from Zhejiang Province Soft Science Research Project (2022C35101).The funders had no role in study design, data collection and analysis, decision to publish, or preparation of the manuscript."

Reviewers' comments:

Reviewer's Responses to Questions

**Comments to the Author**

1. Is the manuscript technically sound, and do the data support the conclusions?

Reviewer #1: No

Reviewer #2: Yes

2. Has the statistical analysis been performed appropriately and rigorously? 

Reviewer #1: Yes

Reviewer #2: Yes

3. Have the authors made all data underlying the findings in their manuscript fully available?

Reviewer #1: Yes

Reviewer #2: Yes

4. Is the manuscript presented in an intelligible fashion and written in standard English?

Reviewer #1: No

Reviewer #2: Yes

5. Review Comments to the Author

Reviewer #1: The topic is interesting, and the author has done a great job in realizing the subject. However, there are few areas on the paper that is still lagging and should be addressed properly:

1.The paper language needs to be improved and proofread. I would suggest to the authors ask an native English speaker, or a professional agency to take on this role.

2. The process of building policy networks needs to be more detailed

3. The sample could have been chosen to be more typical, and it is recommended that additional samples such as the Tokyo Bay Area and the San Francisco Bay Area, which have more mature Bay Area economies

Reviewer #2: This paper is generally well-written and addresses an intriguing and significant topic in energy economics literature. Despite the comprehensiveness and profundity of the research in this work, I have some suggestions for the authors to consider.

Must be discussed these points in the introduction section:

• How can machine-learning algorithms evaluate the urgency and tone of the sustainable development policy texts from China's Bay Area?

• How closely do the policy text networks resemble or deviate from global sustainable development objectives?

• What are the main hubs and operators in China's Bay Area policy text network for sustainable economic growth, and how have they changed over time?

• How do regional variations within the Bay Area of China show up in the network of texts on sustainable economic policy?

• Please do note that in the introduction section, you need to motivate the research area and find the research gap (what this study tries to fill). Currently, from my perspective research area is not written properly to motivate the reader, the research gap is vague and even references are not enough to justify the topic. follow and cite some recent research papers such as " Investigating the interaction effect of urbanization and natural resources on environmental sustainability in Pakistan". 2) "Toward achieving environmental sustainability target in Organization for Economic Cooperation and Development countries: The role of real income, research and development, and transport infrastructure". 3) Linking economic growth and ecological footprint through human capital and biocapacity. 4) . Role of institutions in correcting environmental pollution: An empirical investigation. 5) Combined nonlinear effects of urbanization and economic growth on CO2 emissions in Malaysia. An application of QARDL and KRLS. 6) How do green energy investment, economic policy uncertainty, and natural resources affect greenhouse gas emissions? A Markov-switching equilibrium approach

These points should be discussed in the policy and conclusion

• What ethical ramifications may policy text network analysis employing AI have, particularly in terms of sustainable economic development?

• How does policy text network analysis relate to current qualitative techniques for examining policy impact?

• What are the constraints of utilizing policy text network analysis to assess the effectiveness of policies for sustainable economic development?

6. PLOS authors have the option to publish the peer review history of their article (what does this mean?). If published, this will include your full peer review and any attached files.

Reviewer #1: No

Reviewer #2: No

---

## [Author Response · Author response to Decision Letter 0]

8 Nov 2023

Dear Editor and Reviewers,

Thank you very much for your letter and the comments from the referees about our paper submitted to PLOS one. We have checked the manuscript and revised it arccording to the comments. We submit here the revised manuscript, revised manuscript with track changes, response to reviewers, as well as a list of changes.

If you have any question about this paper, please don't hesitate to let us know.

Sincerely yours,

Huijie Zhou, Shangjia Yu and Pengyue Wu

zhouhuijie@nbu.edu.cn(Huijie Zhou), wupengyue@nbu.edu.cn(Pengyue Wu)

---

## [Decision Letter · Decision Letter 1]

8 Dec 2023

Analyzing the impact of sustainable economic development from the policy text network: Based on the practice of China's Bay area policy

PONE-D-23-23466R1

Dear Dr. Huijie Zhou,

We’re pleased to inform you that your manuscript has been judged scientifically suitable for publication and will be formally accepted for publication once it meets all outstanding technical requirements.

Kind regards,

Kashif Iqbal, Ph.D.,

Academic Editor

PLOS ONE

Additional Editor Comments (optional):

Reviewers' comments:

Reviewer's Responses to Questions

**Comments to the Author**

1. If the authors have adequately addressed your comments raised in a previous round of review and you feel that this manuscript is now acceptable for publication, you may indicate that here to bypass the “Comments to the Author” section, enter your conflict of interest statement in the “Confidential to Editor” section, and submit your "Accept" recommendation.

Reviewer #1: All comments have been addressed

Reviewer #2: All comments have been addressed

2. Is the manuscript technically sound, and do the data support the conclusions?

Reviewer #1: Partly

Reviewer #2: Yes

3. Has the statistical analysis been performed appropriately and rigorously? 

Reviewer #1: Yes

Reviewer #2: Yes

4. Have the authors made all data underlying the findings in their manuscript fully available?

Reviewer #1: No

Reviewer #2: Yes

5. Is the manuscript presented in an intelligible fashion and written in standard English?

Reviewer #1: Yes

Reviewer #2: Yes

6. Review Comments to the Author

Reviewer #1: (No Response)

Reviewer #2: All the comments have been well addressed. The authors have done great work and the paper is accepted for publication

7. PLOS authors have the option to publish the peer review history of their article (what does this mean?). If published, this will include your full peer review and any attached files.

Reviewer #1: No

Reviewer #2: No

---

## [Editor Report · Acceptance letter]

18 Dec 2023

PONE-D-23-23466R1 

PLOS ONE

Dear Dr. Zhou, 

I'm pleased to inform you that your manuscript has been deemed suitable for publication in PLOS ONE. Congratulations! Your manuscript is now being handed over to our production team.

Kind regards, 

on behalf of

Prof.Dr. Kashif Iqbal 

Academic Editor

PLOS ONE